# High-Performance Sample Substrate of Gold Nanoparticle Multilayers for Surface-Assisted Laser Desorption/Ionization Mass Spectrometry

**DOI:** 10.3390/nano9081078

**Published:** 2019-07-27

**Authors:** Yen-Chen Liu, Yi-Hsuan Chang, Yun-Ho Lin, Chien-Chung Liou, Tsung-Rong Kuo

**Affiliations:** 1Graduate Institute of Nanomedicine and Medical Engineering, College of Biomedical Engineering, Taipei Medical University, Taipei 11031, Taiwan; 2Division of Oral Pathology, Department of Dentistry, Taipei Medical University Hospital, Taipei 11031, Taiwan; 3School of Dentistry, College of Oral Medicine, Taipei Medical University, Taipei 11031, Taiwan; 4Department of Chemistry, Tunghai University, Taichung 40704, Taiwan; 5International Ph.D. Program in Biomedical Engineering, College of Biomedical Engineering, Taipei Medical University, Taipei 11031, Taiwan; 6Research Center of Biomedical Device, Taipei Medical University, Taipei 11031, Taiwan

**Keywords:** gold nanoparticles, self-assembly, gold nanoparticle multilayers, sample substrate, photothermal effect, reproducibility

## Abstract

The development of a sample substrate with superior performance for desorption and ionization of analyte is the key issue to ameliorate the quality of mass spectra for measurements of small molecules in surface-assisted laser desorption/ionization mass spectrometry (SALDI-MS). Herein, the homogeneous sample substrate of gold nanoparticle multilayers (AuNPs-ML) with hexagonal lattice was successfully prepared by self-assembly technique. With strong surface plasmon resonance absorption and superior photothermal effect, the sample substrate of AuNPs-ML exhibited high signal sensitivity and low background noise for the detection of model analyte of glucose without additional matrixes in SALDI-MS. Furthermore, compared to merchant matrixes of α-cyano-4-hydroxycinnamic acid (CHCA) and 2,5-dihydroxybenzoic acid (DHB), the sample substrate of AuNPs-ML was demonstrated to ameliorate the quality of mass spectra, including signal strength, background interference and signal/noise (S/N) ratio. The sucrose and tryptophan were also measured to show the extensive applications of AuNPs-ML sample substrate for the detections of small molecules in SALDI-MS. Most importantly, the remarkable reproducibility of glucose mass spectra with relative signal of 7.3% was obtained by the use of AuNPs-ML sample substrate for SALDI-MS. The homogeneous sample substrate of AuNPs-ML greatly improved the quality of mass spectra because of its strong absorption of laser energy, low specific heat, high heat conductivity and extraordinary homogeneity. We believe that AuNPs-ML could be a practical sample substrate for small molecule detection in SALDI-MS.

## 1. Introduction

Surface-assisted laser desorption/ionization mass spectrometry (SALDI-MS) has been extensively utilized for the analyses of small molecules (m/z < 500 Da) such as drugs, pesticides and biomolecules [1,2,3,4,5,6,7,8,9,10,11]. With the applications of nanomaterials as the assisted matrixes, SALDI-MS has shown specific advantages included facile sample fabrication, strong sample signal, less background noise and excellent salt tolerance [12,13,14,15,16,17]. Various nanomaterials such as semiconductor, metal, and metal oxide have been widely applied as superior matrixes due to their intense light absorption, large surface area and high stability, resulting in high signal/noise (S/N) ratio for small molecules [18,19,20,21,22]. Recently, the matrix of CeO_2_-carbon black nanocomposite has been utilized as a substrate in SALDI-MS for the detections of drug doping beverages without pretreating processes of analytes in the fields of forensic toxicologists [23]. Hydrothermal preparation of zinc oxide-reduced graphene oxide hybrid has been developed as a matrix for the detection of both aromatic and aliphatic contaminants on the wafer [24]. Nanostructured mesoporous germanium SALDI substrate has been proved for the measurements of unlawful cocaine in the surroundings, including workplace and roadside [25]. With the use of HgTe nanostructures as a matrix and sucralose as an internal standard, the contents of monosaccharides and disaccharides in honey samples have been measured with short time (<30 min) and high reproducibility (relative standard deviation <15%) without time-consuming sample pretreatment and purification [26]. Nanomaterial-based matrixes have been successfully demonstrated for the detections of many small molecules by SALDI-TOF-MS. However, there are still some crucial issues that need to be addressed such as the increase of signal intensity, decrease of background noise and improvement of sample reproducibility.

For the nanomaterial-based matrixes, gold nanoparticles have shown outstanding performance because of their unique surface plasmon resonance property, low specific heat and high heat conductivity [27,28,29,30]. Gold nanoparticles have shown a wide surface plasmon absorption from ultraviolet to visible light, corresponding to the laser wavelength in SALDI-MS. Furthermore, with intense surface plasmon resonance and photothermal effect, gold nanoparticles can efficiently absorb SALDI-MS laser energy and then transform the laser energy into heat for the increase of desorption and ionization of analytes [31,32,33,34]. Moreover, with low specific heat and high heat conductivity, the temperature of gold nanoparticles can be quickly increased and then the heat produced by gold nanoparticles can be dissipated to the surroundings in order to facilitate the desorption and ionization of analytes [35,36,37]. For example, the matrixes of bare gold nanoparticles have been used to cationize small neutral carbohydrates captured on their surface in order to improve the desorption/ionization efficiency in SALDI-MS [38]. Biomarkers of carcinoid tumors such as tryptophan, 5-hydroxytryptophan, 5-hydroxytryptamine, and 5-hydroxyindole acetic acid have been successfully measured in the urine samples with the applications of gold nanoparticles as the matrixes by SALDI-MS [39]. One-pot synthesis of gold nanoparticles conjugated with dopamine dithiocarbamate as the matrixes have revealed a superior desorption and ionization capability for accurate measurement of low molecular weight analytes including amino acid, drugs and peptides [40]. Gold nanoparticles have been demonstrated as the matrixes for effective ionization of microbial cellular extract ingredients without matrix interference [41]. Applications of gold nanoparticles have been proven as the outstanding matrixes for the analyses of small molecules in SALDI-MS. However, the important crux of low reproducibility of mass spectra still needs to be addressed for future developments of gold nanoparticles as the matrixes.

The matrixes of nanomaterials included gold nanoparticles have suffered from low reproducibility of mass signals due to inhomogeneous distributions of the nanomaterials on sample wells [42,43,44,45]. Recent studies have made a great effort to control the structure and the homogeneity of the assisted nanomaterial matrixes to constitute homogeneous films on the steel plate to improve the reproducibility of mass signals in SALDI-MS [46,47,48,49]. Homogeneous films of nanomaterials have been developed as the SALDI substrates as well as the matrixes to increase the reproducibility of mass spectra. For instance, layer-by-layer graphene oxide/gold nanoparticle thin films that have been prepared as a SALDI substrate to analyze small molecules such as amino acids, saccharide and polypeptides in SALDI-MS with the relative signal, varied less than 15% [50]. Hybrid nanoporous structures of layer-by-layer electrostatic self-assembly of silver nanoparticles and reduced graphene oxide have been employed as a facile platform for high-speed detection of carboxylic acid derivatives such as amino acids, fatty acids, and organic dicarboxylic acids with low matrix interference by SALDI-MS [51]. Gold nanoporous films modified with cysteine have been fabricated and utilized as the SALDI substrates for the analyses of various molecules included amino acids, drug, cyclodextrins, peptides, and polyethylene glycols with relative signal less than 10% [52]. Langmuir-Blodgett films of shape-controlled silver nanocrystals with large area have been manufactured as a matrix-free sample plate for glucose detection with relative signal of 5.7% in SALDI-MS [53]. Although the utilizations of nanomaterial-based homogeneous films as the SALDI substrates have revealed superior reproducibility of mass signal, the practical uses of those thin films are still restricted due to the lack of a simple approach to fabricate large-scale and high-quality thin films for SALDI-MS detections.

In this study, an amiable design was employed to self-assemble gold nanoparticles into AuNPs-ML as shown in Figure 1. With a highly oriented structure, the homogenous AuNPs-ML was utilized as a high-performance substrate for SALDI-MS measurement. With various layers, the optimal status of AuNPs-ML sample substrate for measurement of small molecule of glucose was demonstrated without extra matrix in SALDI-MS. Furthermore, the efficiency of AuNPs-ML for desorption/ionization of glucose was investigated compared with merchant matrixes, including CHCA and DHB. To further examine the performance of a sample substrate of AuNPs-ML, small molecules of sucrose and tryptophan were measured in SALDI-MS measurements. Moreover, the reproducibility of mass spectra of glucose was studied with the uses of AuNPs-ML sample substrates in SALDI-MS.

## 2. Materials and Methods

### 2.1. Chemicals

Gold acetate (99.9%) was purchased from Alfa Aesar (Ward Hill, MA, USA). 1,2-hexadecandiol (90%), ethanol (95%), α-cyano-4-hydroxy-cinnamic acid (CHCA), and 2,5-dihydroxybenzoic acid (DHB) were purchased from Sigma-Aldrich (St. Louis, MO, USA). 1-octadecene (ODE), oleic acid (OA), oleylamine (OLA), and acetonitrile (99.8%) were purchased from Acros Organics (Houston, TX, USA). Toluene (HPLC grade) was purchased from Fisher Scientific (Pittsburgh, PA, USA). Hexane (HPLC grade) was purchased from Duksan (Kyunggi, South Korea). 1,2-ethandithiol (EDT) was purchased from Fluka (St. Gallen, Swiss).

### 2.2. Preparation of Gold Nanoparticles

Gold nanoparticles were prepared via one-pot synthesis according to the previous study with some modification [54]. First, 20 mL of ODE, 2 mmol of OLA, 2 mmol of OA, 0.5 mmol of gold acetate, and 4 mmol of 1,2-hexadecanediol were added to a three-necked round bottle flask and then degassed under vacuum for 1 h at 130 °C. After that, under the nitrogen atmosphere, the mixture was heated to 220 °C and further kept at 220 °C for 1 h for the synthesis of gold nanoparticles. After 1 h, the gold nanoparticles were obtained. For purification, the gold nanoparticle solution was precipitated by ethanol and then centrifuged at 8000 rpm for 30 min. After removal of supernatant, the precipitation of gold nanoparticles was redispersed in hexane. The washing steps were repeated several times. The gold nanoparticles were finally dispersed in hexane for further experiments.

### 2.3. Fabrication of Self-Assembled Monolayer of Gold Nanoparticles

To fabricate the self-assembled monolayer (SAM) of gold nanoparticles, stock solution of gold nanoparticles was first prepared by adding 700 μL of gold nanoparticle solution into 300 μL of toluene. After, a glass petri dish (10 cm × 1.5 cm) was applied as trough to fabricate SAM of gold nanoparticles. The trough of glass petri dish was sequentially cleaned by soap water, deionized water, and ethanol and then dried in an oven at 70 °C. Before fabrication of SAM of gold nanoparticles, 75 mL of deionized water was added to the trough of glass petri dish. Afterward, 1 mL of stock solution of gold nanoparticles was added dropwise on the surface of deionized water in the trough by a pipette. A cover with a hole (3 cm^2^) at the edge was used to cover the trough of glass petri dish with gold nanoparticles. Twenty-four hours after the evaporation of hexane and toluene, the SAM of gold nanoparticles was obtained on the surface of deionized water. To prepare sample substrate for SALDI-MS applications, the steel plate (1 cm × 1 cm) was slowly immersed into the trough of glass petri dish. Following, the steel plate was slowly pulled out of the sub-phase to transfer SAM of gold nanoparticles.

### 2.4. Gold Nanoparticle Multilayers as the Sample Substrates for SALDI-MS Measurements

For the fabrication of sample substrate of AuNPs-ML, SAM of gold nanoparticles was repeatedly transferred onto the steel plate with the target layers. To increase the surface hydrophilicity of AuNPs-ML, sample substrate of AuNPs-ML was immersed into EDT (2 mM) in acetonitrile for 90 s. After the treatment of AuNPs-ML by 1,2-ethandithiol, 1 μL of analyte solution was straightly loaded onto the sample substrate of AuNPs-ML and then the sample substrate was dried in a vacuum oven. For SALDI-MS application, the sample substrate of AuNPs-ML with analyte was analyzed without adding any additional matrix. Moreover, the analytes were also analyzed with the merchant matrixes, which included CHCA (10 mg/mL) and DHB (20 mg/mL), to examine their efficiencies of SALDI-MS measurements. First, 1 μL of CHCA and 1 μL of DHB were separately loaded onto steel plates and then dried in a vacuum oven. Second, 1 μL of analyte solution was then deposited onto the merchant matrix of CHCA or DHB and then dried in a vacuum oven. Finally, the steel plate with merchant matrix and analyte was further analyzed by SALDI-MS.

### 2.5. SALDI-MS Measurements

Mass spectrometry (microflex, Bruker, Hamburg, Germany) was operated in positive-ion reflector mode using a 1.25 m flight tube. The nitrogen laser with wavelength at 337 nm was utilized to illuminate analyte on the sample substrate with pulse duration of 3 ns and frequency of 10 Hz. During the nitrogen laser irradiation for desorption and ionization of analyte, the ions of analyte were maintained with a delayed extraction period of 100 ns, and further accelerated through the linear time-of-flight reflection before going into the mass analyzer. The accelerating voltages were applied in the range from +20 to −20 kV. In the process of SALDI-MS detection, the laser power was tuned to be 130 μJ to acquire the best quality of mass spectra with high signal intensity, minimal interference and excellent S/N ratio. Each spectrum of analyte was generated by an average of 100 laser pulses. All experiments were repeated at least three times to prove the reproducibility of the mass spectra.

## 3. Results and Discussion

### 3.1. Structural and Optical Properties of Gold Nanoparticles

The gold nanoparticles prepared via one-pot synthesis were first characterized by transmission electron microscopy (TEM, Hitachi HT7700, Tokyo, Japan) and UV-Vis absorption spectroscopy (JASCO V-770 with ISN 923 Integrating Sphere, Easton, MD, USA) to investigate their structural and optical properties. In TEM image of Figure 2a, the gold nanoparticles revealed nearly spherical shape and uniform size. In Figure 2b, the histogram of size distributions of gold nanoparticle was statistically counted on 100 nanoparticles in TEM image of Figure 2a. Gaussian fitting curve of the histogram of size distributions for gold nanoparticle was also simulated as shown in Figure 2b. After fitting a Gaussian curve to the histogram of size distributions for gold nanoparticle, the average size of gold nanoparticles was calculated to be 8.2 nm. Moreover, the gold nanoparticles exhibited an obvious surface plasmon resonance absorption at ~520 nm, as shown in absorption spectra of Figure 2c. Furthermore, gold nanoparticles have shown a broad absorption at wavelength of 337 nm corresponding to the emission of nitrogen laser applied in SALDI-MS measurements. Taking advantages of strong surface plasmon resonance absorption and superior photothermal effect, the uniform gold nanoparticles can be a promising candidate for the preparation of homogeneous sample substrate to improve desorption/ionization of analytes (*m*/*z* < 500 Da) by SALDI-MS.

### 3.2. Characterizations of Self-Assembled Monolayer of Gold Nanoparticles

After preparation of gold nanoparticles, self-assembly technique was utilized to fabricate monolayer film of gold nanoparticles. The SAM of gold nanoparticles was transferred onto copper grid and then characterized by TEM. As shown in Figure 3a, gold nanoparticles were self-assembled into two-dimensional monolayer film with hexagonal lattice. The homogeneous SAM of gold nanoparticles with a highly oriented structure was obtained because of thermodynamic equilibrium [55]. Furthermore, the absorption spectra of SAM of gold nanoparticles was measured by UV-Vis absorption spectroscopy. As shown in the absorption spectrum of Figure 3b, the surface plasmon resonance absorption of SAM of gold nanoparticles was extremely affected by the plasmon coupling between gold nanoparticles [56,57]. The SAM of gold nanoparticles exhibited strong and broad absorption band, from 300 nm to 700 nm. Therefore, with superior homogeneity and strong absorption at a wavelength of 337 nm, the SAM of gold nanoparticles could be applied as a potential sample substrate to improve spectral quality and reproducibility in SALDI-MS. Most importantly, for analyses of biological samples, surface hydrophilicity is a key parameter for development of the sample substrate in SALDI-MS. To increase the surface hydrophilicity, the SAM of gold nanoparticles was treated by EDT [58]. Before and after treatment of EDT, the surface hydrophilicity of SAM of gold nanoparticles was measured by contact angle. As shown in Figure 3c,d, the contact angles were respectively measured to be 115° and 75°, before and after EDT treatment. The results of contact angle measurements indicated that the surface hydrophilicity of sample substrate of SAM of gold nanoparticles was successfully increased by EDT treatment.

### 3.3. Investigation of Gold Nanoparticle Multilayers as the Sample Substrates in SALDI-MS

For the development of sample substrate of AuNPs-ML, SAM of gold nanoparticles was repeatedly transferred onto the steel plate. To investigate the performance of the sample substrate, AuNPs-ML with 10, 15, 20, and 25 layers (AuNPs-ML10, AuNPs-ML15, AuNPs-ML20, and AuNPs-ML25) were respectively applied as the sample substrates to analyze glucose (10^−3^ M) in MALDI-MS. The analyte of glucose is often applied for the first examination of the capability of SALDI-MS substrate for small molecule measurement. The mass spectrum of glucose (10^−3^ M) was first detected on the steel plate in SALDI-MS as shown in Appendix A. There is no significant glucose signal with the application of steel plate in SALDI-MS measurement. Moreover, the mass spectrum of sample substrate of AuNPs-ML15 was measured to examine the background signals in SALDI-MS, as shown in Appendix A. The background signals of AuNPs-ML15 also showed no significant signal. With the uses of AuNPs-ML as the sample substrates, the mass spectra of glucose were successfully obtained as shown in Figure 4. By using the sample substrates of AuNPs-ML, the glucose was detected as the [Glucose+Na]^+^ ion with high signal intensity in SALDI-MS. With the increases of AuNPs-ML layers from 5 to 10, the intensity of glucose signal was increased and then achieved the maximal intensity with the use of sample substrate of AuNPs-ML15 in SALDI-MS. The maximal intensity of glucose signal obtained with AuNPs-ML15 was attributed to the fact that the sample substrate of AuNPs-ML15 exhibited higher absorption coefficient at 337 nm than that of AuNPs-ML10. The absorption coefficients at 337 nm for AuNPs-ML10 and AuNPs-ML15 were 0.46 and 0.57, respectively. AuNPs-ML15 with higher absorption coefficient at 337 nm can absorb more nitrogen laser and then transform into heat to facilitate desorption/ionization of glucose during SALDI-MS measurement. However, the intensity of glucose signal decreased when the AuNPs-ML layers were increased more than 15 layers. The decrease of signal intensity was evidently shown from AuNPs-ML20 to AuNPs-ML25. The results can be ascribed to the fact that the analyte of glucose penetrated into the sample substrate of AuNPs-ML, resulting in the difficulty for desorption and ionization of glucose in the bottom layers in SALDI-MS. Therefore, the optimal sample substrate of AuNPs-ML15 was applied for the following SALDI-MS measurements.

### 3.4. Merchant Matrixes and AuNPs-ML for SALDI-MS Measurements

To investigate the performance of AuNPs-ML sample substrate, merchant matrixes included CHCA and DHB were employed to measure glucose (10^−3^ M) in SALDI-MS. As shown in Figure 5, the interference coming from the matrix of CHCA was extremely strong for glucose detection in SALSI-MS. With the additive of merchant assisted matrix of DHB, the weak intensity of glucose signal and the low signal-to-noise ratio were obtained in comparison with the sample substrate of AuNPs-ML15. Overall, the sample substrate of AuNPs-ML15 exhibited the highest intensity of glucose signal, the lowest background noise and the excellent S/N ratio compared to that of merchant matrixes of CHCA and DHB. The results demonstrated that AuNPs-ML15 has revealed the great potential for the use as a sample substrate for the detections of small molecules without additional matrix in SALDI-MS.

### 3.5. Detections of Sucrose and Tryptophan

To further examine the performance of sample substrate of AuNPs-ML15, small molecules of sucrose (10^−3^ M) and tryptophan (10^−3^ M) were applied as analytes in SALDI-MS measurements. As shown in Figure 6, the sucrose was detected as the [sucrose+H]^+^ and [sucrose+Na]^+^ ions and the tryptophan was measured as the [tryptophan+H]^+^ ion. Both of the mass spectra of sucrose and tryptophan exhibited strong signal intensities and low background signals. The results indicated that very few fragment ions of the analytes can be detected with the use of sample substrate of AuNPs-ML15 in SALDI-MS. Furthermore, the sample substrate of AuNPs-ML15 showed no significant interference for the measurements of sucrose and tryptophan in SALDI-MS. Overall, AuNPs-ML15 was demonstrated as the sample substrate to obtain strong signal intensity and low background signal for small molecules in SALDI-MS.

### 3.6. Signal Reproducibility of AuNPs-ML Sample Substrate in SALDI-MS

To realize the practical application, the reproducibility of mass spectra is a common issue that needs to be improved for the enhancement of reliability in SALDI-MS detection. Herein, AuNPs-ML-15 was applied as a homogeneous sample substrate to improve the reproducibility of mass spectra. As shown in Figure 7, the relative signal of [Glucose+Na]^+^ was estimated to be 7.3% with the application of AuNPs-ML-15 sample substrate in SALDI-MS. Compared to merchant glucometer with the error margin of relative signal of 15%, the sample substrate of AuNPs-ML-15 can provide an accurate and precise approach for glucose measurement in SALDI-MS. The excellent reproducibility of mass spectra could be attributed to the fact that the sample substrate of AuNPs-ML-15 exhibited strong absorption at wavelength of 337 nm, low specific heat, excellent heat conductivity, and superior homogeneity. With the strong absorption at wavelength of 337 nm, the sample substrate of AuNPs-ML-15 can absorb more laser energy and then transform into heat by photothermal effect for the increase of desorption and ionization of analyte in SALDI-MS. Furthermore, the temperature of AuNPs-ML-15 sample substrate can be rapidly increased to improve desorption and ionization of analyte because of its low specific heat. With the excellent heat conductivity, the sample substrate of AuNPs-ML-15 can swiftly transfer heat to analyte to induce desorption and ionization. Most importantly, the superior homogeneity of AuNPs-ML-15 sample substrate can homogeneously transfer the heat to improve desorption/ionization of analyte in SALDI-MS. Therefore, the limit-of-detection reached to 1 μM for the glucose measurement with the application of AuNPs-ML15 as the sample substrate in SALDI-MS. To sum up, the analysis of analyte with the sample substrate of AuNPs-ML can provide a practical approach by SALDI-MS.

## 4. Conclusions

The homogeneous AuNPs-ML with hexagonal lattice was successfully prepared and served as a sample substrate in SALDI-MS measurement. The sample substrate of AuNPs-ML15 revealed the maximal signal intensity of glucose compared to AuNPs-ML10, AuNPs-ML20, and AuNPs-ML25 due to the increase of desorption and ionization of glucose in SALDI-MS. AuNPs-ML15 sample substrate also revealed the best quality of glucose mass spectra including the highest signal intensity, the lowest background noise, and the excellent S/N ratio in comparison with merchant matrixes of CHCA and DHB. Various analytes including sucrose and tryptophan were simply measured by the uses of AuNPs-ML15 sample substrate in SALDI-MS. Moreover, the sample substrate of AuNPs-ML-15 with strong absorption at wavelength of 337 nm, low specific heat, excellent heat conductivity, and superior homogeneity can provide an accurate and precise approach for the measurements of small molecules in SALDI-MS.

## Figures and Tables

**Figure 1 nanomaterials-09-01078-f001:**
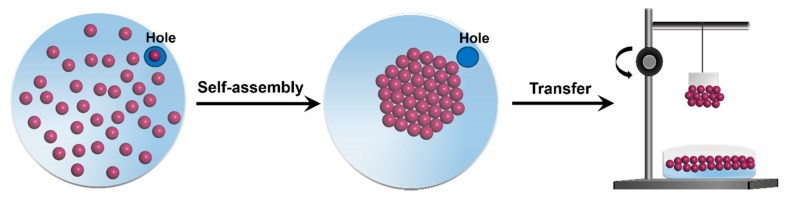
Schematic illustration of self-assembly of gold nanoparticles and fabrication of AuNPs-ML sample substrate. Briefly, stock solution of gold nanoparticles was added dropwise on the surface of deionized water in the trough. Afterward, a cover with a hole at the edge was covered onto the trough. After evaporation for 24 h, the self-assembled gold nanoparticles were obtained on the surface of deionized water. A steel plate was slowly pulled out of the sub-phase to transfer self-assembled gold nanoparticles.

**Figure 2 nanomaterials-09-01078-f002:**
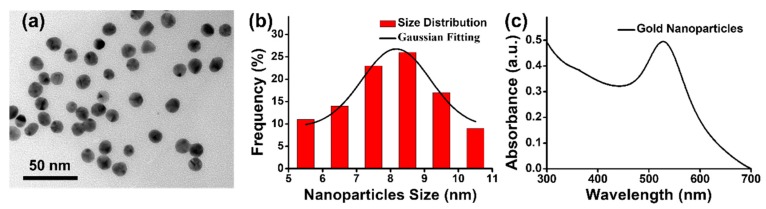
(**a**) TEM image of gold nanoparticles. (**b**) Histogram of gold nanoparticle size distributions and their Gaussian fitting. (**c**) UV-Vis absorption spectra of gold nanoparticle solution.

**Figure 3 nanomaterials-09-01078-f003:**
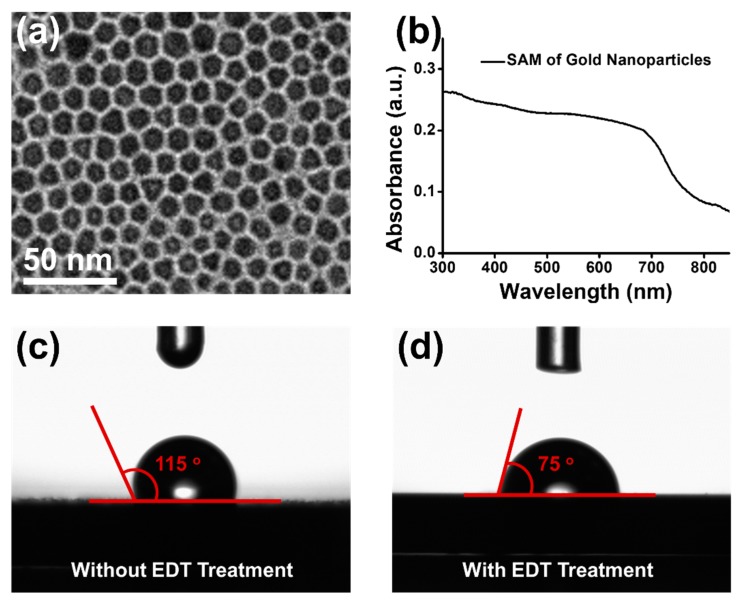
(**a**) TEM image of SAM of gold nanoparticles. (**b**) UV-Vis absorption spectrum of SAM of gold nanoparticles. Contact angles of SAM of gold nanoparticles before (**c**) and after (**d**) EDT treatment.

**Figure 4 nanomaterials-09-01078-f004:**
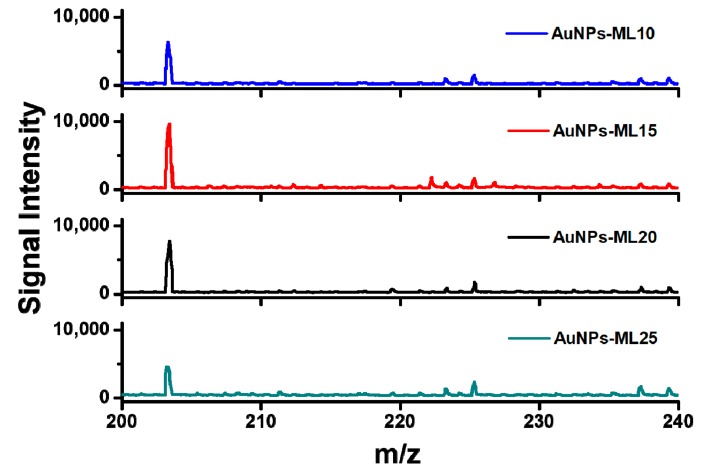
Mass spectra of glucose with the application of AuNPs-ML as the sample substrates in SALDI-MS. The concentration of glucose was 10^−3^ M. Peak identity: m/z 203.26, [Glucose+Na]^+^.

**Figure 5 nanomaterials-09-01078-f005:**
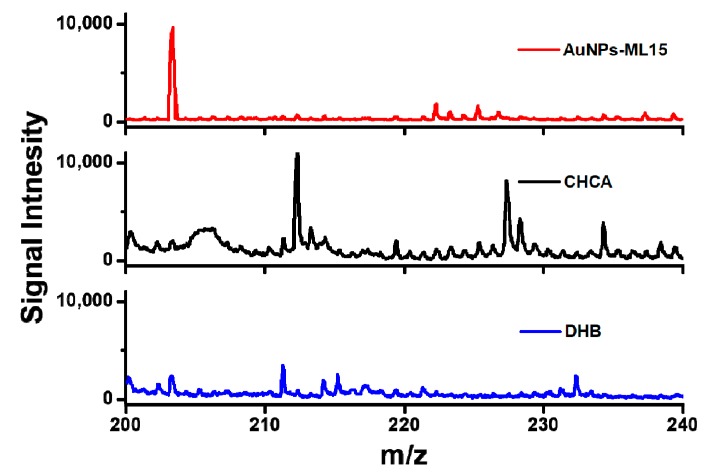
Mass spectra of glucose using AuNPs-ML15 as the sample substrate and merchant matrixes of CHCA and DHB. Peak identity: m/z 203.26, [Glucose+Na]^+^.

**Figure 6 nanomaterials-09-01078-f006:**
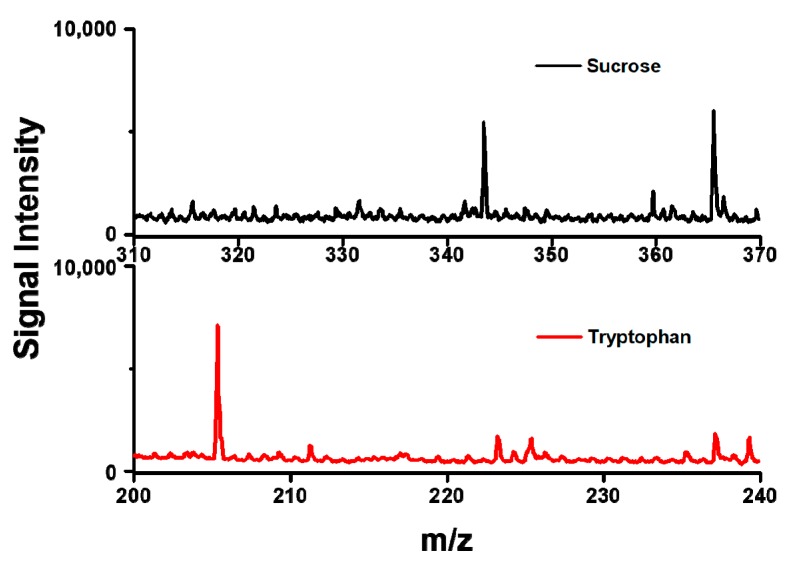
Mass spectra of sucrose and tryptophan using AuNPs-ML15 as the sample substrate in SALDI-MS. Peak identity: m/z 343.46, [sucrose+H]^+^; m/z 365.46, [sucrose+Na]^+^; 205.25 m/z, [tryptophan+H]^+^.

**Figure 7 nanomaterials-09-01078-f007:**
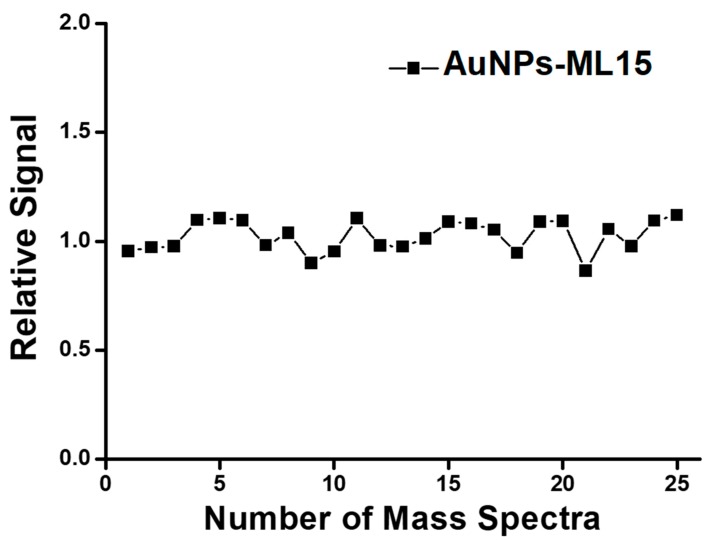
The relative signal of glucose was measured and calculated with the sample substrate of AuNPs-ML-15 in SALDI-MS.

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
