# Peer review of "High-Performance Sample Substrate of Gold Nanoparticle Multilayers for Surface-Assisted Laser Desorption/Ionization Mass Spectrometry"

_nanomaterials, 2019, doi:10.3390/nano9081078_

Reviewer 1 Report

Nanomaterials-554385

This paper presents homogeneous AuNPs-ML with hexagonal lattice successfully prepared, served as a sample substrate in SALDI-MS measurement. Results indicate that the sample substrate of AuNPs-ML-15 can provide an accurate and precise approach for the measurements of small molecules in SALDI-MS. Although in the referee´s opinion the work is interesting this paper does not have the high standards needed for publication in Nanomaterials. Some references in the introduction are very old and should be updated. The physicochemical study is not complete enough. For example, the gold nanoparticle solution was precipitated by ethanol and then centrifuged at 8000 rpm for 30 min. Why at 8000 rpm when usually you need more than 12000?. This point must be clarified or references given. The histogram in Fig. 2 should be improved. Authors have only considered about one hundred gold nanoparticles. No monodispersity data are indicated, although it seems that they are, nor of sphericity percentages. In summary, although the experiments are correct and the results consistent, the manuscript can be improved before it deserves to be published.

Reviewer 2 Report

In the present paper, the authors report on the fabrication of sample substrates for measurements of small molecules in surface-assisted laser desorption/ionization mass spectrometry (SALDI-MS). In particular, they prepared homogeneous substrates of gold nanoparticle multilayers (AuNPs-ML) with hexagonal lattice by self-assembly technique. The optical and structural properties of substrates were characterized and glucose, sucrose and tryptophan were used for testing the performances of prepared substrates.

The manuscript is well-organized and clearly written. Some points indicated below should be considered before the work can be accepted for publication in Nanomaterials Journal.

List of points to be addressed

1)      The Introduction has to be more focused on the specific argument of the manuscript, the number of cited references is certainly excessive. Please try to select those most relevant to the subject of the paper.

2)      Please check the size of the hole cited in 2.3. Fabrication of Self-Assembled Monolayer of Gold Nanoparticles paragraph. It seems too large to me. In the same paragraph, please better explained where the steel plate is initially positioned.

3)      Please correct the details on the mass spectrometer at the beginning of 2.5 paragraph. Also, indicate the average laser power used during the experiments.

4)      As far as concerns the surface plasmon resonance absorption at ~ 520 nm, it doesn’t seem to be in agreement with the size of the calculated average size of gold nanoparticles  (8.2 nm). A surface plasmonic resonance at around 520 nm is generally associated with larger particles. See for example Haiss et al Anal. Chem. 2007, 79, 4215-4221 and similar papers. The authors have to explain their result.

5)      As far as concerns the spectra in Figure 3b, it is largely different from those reported in the Refs. 61 and 62 cited by the authors. What can be the reason for this?

6)      The authors affirm …Moreover, the mass spectrum of sample substrate of AuNPs-ML15 was measured to examine the background signals in SALDI-MS as shown in Figure S2. The background signals of AuNPs-ML15 also showed no significant glucose signal. If Figure S2 is related to the background signals, it is obvious that there no glucose signal. Please rewrite these sentences.

7)      It would be very useful if the authors add some comments about the specificity and the limit-of-detection that can be obtained by using the proposed substrates in SALDI-MS measurements. Also, a comparison with other not commercial substrates would be interesting for the reader.

8) Please check the manuscript for some misprints. In addition, a careful revision of the English language would be useful for eliminating some minor errors.

Author Response

Round  2

Reviewer 1 Report

Authors have considered all the points. The manuscript merits to be published.